# Pre-warming before general anesthesia with isoflurane delays the onset of hypothermia in rats

**Maxime Rufiange[1,2], Vivian S. Y. Leung[1,2], Keith Simpson[3], Daniel S. J. Pang[1,2]***

**1** Faculty of Veterinary Medicine, Department of Clinical Sciences, Université de Montréal, Saint-Hyacinthe, QC, Canada, **2** Faculty of Veterinary Medicine, Groupe de Recherche de Pharmacologie Animale du Québec (GREPAQ), Université de Montréal, Saint-Hyacinthe, QC, Canada, **3** Vetronic Services Ltd, Abbotskerswell, England, United Kingdom

* danielpang17@hotmail.com

**Data Availability Statement:** Data supporting the results are available in an electronic repository: Pang, Daniel, 2020, "Pre-warming and perianesthetic hypothermia_01", https://doi.org/10.

## Abstract

General anesthesia causes hypothermia by impairing normal thermoregulatory mechanisms. When using inhalational anesthetic agents, Redistribution of warm blood from the core to the periphery is the primary mechanism in the development of hypothermia and begins following induction of anesthesia. Raising skin temperature before anesthesia reduces the temperature gradient between core and periphery, decreasing the transfer of heat. This prospective, crossover study (n = 17 adult male and female SD rats) compared three treatment groups: PW1% (pre-warming to increase core temperature 1% over baseline), PW40 (pre-warming to increase core temperature to 40˚C) and NW (no warming). The PW1% group was completed first to ensure tolerance of pre-warming. Treatment order was then randomized and alternated after a washout period. Once target temperature was achieved, anesthesia was induced and maintained with isoflurane in oxygen without further external temperature support. Pre-warming was effective at delaying the onset of hypothermia, with a significant difference between PW1% (12.4 minutes) and PW40 (19.3 minutes, p = 0.0044 (95%CI -12 to -2.2), PW40 and NW (7.1 minutes, p < 0.0001 (95%CI 8.1 to 16.0) and PW1% and NW (p = 0.003, 95%CI 1.8 to 8.7). The rate of heat loss in the pre-warmed groups exceed that of the NW group: PW1% versus NW (p = 0.005, 95%CI 0.004 to 0.027), PW40 *versus* NW (p < 0.0001, 95%CI 0.014 to 0.036) and PW1% *versus* PW40 (p = 0.07, 95%CI -0.021 to 0.00066). Pre-warming alone confers a protective effect against hypothermia during volatile anesthesia; however, longer duration procedures would require additional heating support.

## Introduction

Hypothermia remains a common complication encountered in both human and veterinary anesthesia [1–5]. Heat loss during general anesthesia is affected by various patient and environmental factors. Those related to the patient include severity of disease, and intervention planned (e.g. open body cavities) [3, 4]. Factors related to the environment include exposure

7910/DVN/GCIXJ3, Harvard Dataverse, V1, UNF:6: KaxaaRzSaksvTc+8qIe18Q== [fileUNF]"

**Funding:** This work was supported by Natural Sciences and Engineering Research Council (NSERC) Discovery Grant (ID: 424022-2013; DSJP), Fondation Lévesque (DSJP), and Vetronic Services Ltd. Author KS received support in the form of a salary from Vetronic Services Ltd. The specific role of this author is articulated in the 'author contributions' section. The funders had no role in study design, data collection and analysis, decision to publish, or preparation of the manuscript.

**Competing interests:** The authors have read the journal's policy and have the following conflicts: Author KS is a director at Vetronic Services Ltd., the company that designed and built the heating unit. There are no patents, products in development, or marketed products to declare. This does not alter our adherence to all the PLOS ONE policies on sharing data and materials.

to fluids and surfaces at temperatures below core body temperature and continual circulation of cool air in the environment [6]. Critically, though these factors contribute to peri-anesthetic hypothermia, the most important mechanism of hypothermia during general anesthesia is the redistribution of warm blood from the core to the periphery [7]. This explains why hypothermia begins so rapidly after induction of general anesthesia (before surgery begins) and the difficulty in its prevention or reversal [8, 9].

Body temperature is considered a vital sign and hypothermia can have important adverse effects. In humans, a small decrease in core temperature, as little as 1˚C, is associated with prolonged recovery and hospitalisation, increased surgical site infection and contributes to postoperative pain [10–12]. While the known consequences of hypothermia in the veterinary literature are currently limited, delayed recovery from anesthesia has been shown in both dogs and rats [13, 14].

In mammals, core temperature is normally tightly regulated within a narrow range, the inter-threshold range, that spans ± 0.3˚C. General anesthesia impairs thermoregulation through depression of the hypothalamus, the major thermoregulatory center in the brain. As a result, the inter-threshold range increases 10–20 fold, allowing core body temperature to decrease substantially before corrective measures (vasoconstriction, arterio-venous shunting) begin. Depression of thermoregulation in addition to vasodilation induced by many anesthetic agents allows heat to flow down the temperature gradient from the core to peripheral tissues [15, 16]. In general, core temperature follows a distinct pattern during general anesthesia that consists of three phases: 1) redistribution of heat from the core to the periphery, which accounts for approximately 80% of hypothermia during the first hour of anesthesia, 2) a further decrease in core temperature as heat loss exceeds metabolic heat production in the subsequent 2–3 hours and 3) achieving a plateau in temperature over 3–4 hours as core temperature falls low enough for vasoconstriction to occur and reduce metabolic heat loss to the periphery [7, 17].

Understanding the mechanism of hypothermia during anesthesia has led to the successful practice of pre-warming human patients before induction of anesthesia [18]. The goal of pre-warming is too raise the temperature of the periphery so that the temperature gradient with the core is lessened, thereby delaying the decrease in core temperature as thermoregulatory mechanisms are depressed [19]. Previous work has shown potential for pre-warming to be effective in rodents [20].

The primary objective of this study was to assess different pre-warming temperature regimens on core temperature during general anesthesia. We hypothesized that pre-warming animals before induction of general anesthesia would delay the onset of hypothermia. A secondary objective was to compare the accuracy of different temperature measurement sites to core temperature (telemetric capsules implanted in the abdomen).

## Materials and methods

### Animals

Adult female (n = 10) and male (n = 7) CD Sprague–Dawley rats were obtained from a commercial supplier (Charles River Laboratories, Senneville, QC, Canada). Rats weighed 308–412 g (females; age 19–27 weeks old) and 220–576 g (males; age 7–11 weeks old) at the start of the experiment.

### Ethics statement

Study review and approval was provided by the local animal care and use committee of the Université de Montréal (protocol ID 18-Rech-1947), operating under the auspices of the Canadian Council on Animal Care.

Rats were acclimatized to the environment (warming chamber) and experimenter (MR) for 7 days before the experiment. Rats were considered habituated when they readily accepted a treat offered by hand while in the anesthesia induction box. Rats were pair housed in a plastic cage (45 [l] x 24 [w] x 20 [h] cm) with wood chip and shredded paper bedding and a plastic tube for enrichment. The housing environment was controlled: 14h/10h light/dark cycle (lights on at 06:00), temperature (22°C) and humidity (20–25%). Food (Rodent laboratory chow 5075, Charles River Breeding Laboratories, St-Constant, Quebec, Canada) and tap water were provided *ad libitum*. Small treats were also offered *ad hoc* during the project (Supreme Mini-Treats™, Very berry flavor, Bio-Serv, Flemington, NJ 08822, USA; Veggie-Bites™, Bio-Serv, Flemington, NJ 08822, USA; Fruit Crunchies, Bio-Serv, Flemington, NJ 08822, USA).

The project had two phases: 1. Temperature capsule instrumentation surgery and 2. Pre-warming temperature experiment. Sample size was determined *a priori* with an alpha level of 0.05 and power of 90% (G*Power 3.1.9.2, Germany). The target mean difference was 0.5°C in core temperature with a standard deviation of 0.4°C. This was based on the results of a similar project, giving an estimated sample size of 15 rats per treatment group [20].

## Telemetric temperature capsule implantation

On the day of surgery, telemetry capsules (Anipill temperature sensor; Aniview system®, Bodycap, Hérouville-Saint-Clair, France) were activated and accuracy confirmed by immersion in water baths at 35°C and 37°C: bath temperature was checked with a calibrated infrared thermometer (Fluke infrared thermometer 561, Fluke Corporation, Everett, WA, USA; calibrated at 30°C, 45°C and 60°C with an accuracy of +/- 0.1°C). Temperature capsules were sterilised (chlorhexidine gluconate 0.05% immersion for 30 minutes) and rinsed with sterile saline (0.9% NaCl) before implantation.

All surgeries were completed between 17:00 and 20:00. Approximately 30 minutes before surgery, each rat was given meloxicam (2 mg/kg SC, Metacam, 5 mg/mL; Boehringer Ingelheim Vetmedica, Inc, St Joseph, MO, USA) and buprenorphine (0.03 mg/kg SC, Vetergesic, 0.3 mg/mL; Champion Alstoe, Whitby, ON, Canada). Rats were anesthetized individually in an induction chamber (25.7 [l] x 11 [w] x 10.7 [h] cm; Small box, Harvard apparatus, Holliston, Massachusetts, USA) and the isoflurane vaporizer dial set at 5% in 1 L/min of oxygen until loss of the righting reflex, at which time the rat was removed from the chamber and placed in dorsal recumbency on a heat pad (16 × 38 cm; Stoelting Rodent Warmer with Cage Heating Pad, Stoelting Corporation, Wood Dale, IL) with an output maintained at approximately 37°C. General anesthesia was maintained via nose cone with the isoflurane vaporizer set at approximately 1.75%, carried in 1 L/min oxygen.

Fur was clipped from the xiphoid process to the pubis and the skin was cleaned with alcohol and chlorhexidine. A celiotomy was performed with a 15 mm incision, beginning immediately caudal to the umbilicus. The temperature capsule was positioned freely in the peritoneal cavity and the surgical incision closed in two layers. At completion of surgery, the vaporizer was turned off and the rat allowed to recover with 1 L/min of oxygen on the heating pad. The rat was returned to its home cage following return of sternal recumbency. Meloxicam (2mg/kg SC) was administered 24 and 48 hours post-operatively and a food supplement (DietGel Recovery; Clear H2O, Portland, ME, USA) provided in addition to food for the following 7 days. Only rats displaying a positive weight gain proceeded to the second (temperature experiment) phase. The exclusion criteria from the experiment in the post-operative period consisted of: weight loss, telemetric implant failure, lethargy, pain/infection or complication at the surgical site.

## Pre-warming temperature experiment

The pre-warming experiment was conducted 7 days after capsule instrumentation. A prospective cross-over study was conducted, with animals receiving 3 treatments. Treatment 1 (PW1%): pre-warming to a target of 1% increase in core (capsule) body temperature from baseline. Treatment 2 (PW40): pre-warming to a target core temperature of 40°C. Treatment 3 (NW): no pre-warming control group. A core temperature was established for each animal by averaging temperatures recorded between 08:00 and 18:00 the day before the temperature experiment. From this, each rat's individual hypothermia threshold was determined (mean core temperature minus two standard deviations) and used to identify time to hypothermia. [20] Baseline core temperature from all rats were pooled to facilitate general comparisons between treatments. The hypothermia threshold was determined in the same way for pooled data. The PW1% treatment was performed first as a proof of concept and to ensure there were no adverse behavioral effects of warming before randomising treatment order (www.random.org) to the PW40 and NW treatments. A washout period of at least 5 days was allowed between experiments. The study design and single experimenter (MR) during data collection precluded blinding to treatment. Rats were video-recorded when in the warming chamber (25.7 [l] x 11 [w] x 10.7 [h] cm) for all treatment groups and videos reviewed by an observer blinded to treatment (VL) for signs of behaviors associated with potential distress. Behavioral signs were assessed at two timepoints: 1) the first three minutes after the rats were placed in the chamber and 2) last three minutes (before isoflurane was started). The presence of the following behaviors were recorded: pawing or digging, open mouth breathing, abnormal posture (i.e. hunched back or head pressed into corner), audible vocalisations, chromodacryorrhea and rearing [21, 22]. The direction faced (towards or away from the heat source) was also assessed.

Criteria to withdraw an animal from the temperature experiment were: cutaneous thermal injury and a core temperature < 27°C or > 41°C.

Experiments were conducted between 09:00–17:00. Core temperature was recorded every 2.5 minutes in all treatment groups. The following proxy temperatures were also monitored every 5 minutes in all groups: 1. lateral tail base, 2. fur temperature (at the xyphoid process) and 3. rectal temperature (rectal thermometer inserted 6 cm into rectum, Physio Logic Accuflex Pro, Model 16–639; AMG Medical, Montreal, QC, Canada). Rectal thermometer accuracy was confirmed as described for the telemetry capsules and a correction factor applied as necessary. Proxy temperatures were recorded from the loss of righting reflex, as soon as rats were taken out of the warming chamber, until core temperature achieved a nadir of approximately 34°C. Additionally, skin temperature at the level of the elbow and knee (right thoracic and pelvic limb, respectively) was measured with the infrared thermometer just before entry and as soon as rats were taken out the warming chamber.

## Warming chamber heating unit

The warming chamber heating unit (Vetronic Services Ltd, England) consisted of an in-line electrically heated device and an electronic controller. Located within a 10 cm pipe fitted with 22 mm male connectors, four heating coils provide heat to the fresh gas supplied directly from the fresh gas outlet of anaesthetic machine. Temperature sensors within the heating unit provide information on the exit temperature of the fresh gas. By means of a microprocessor, the heating effect was varied to maintain a constant exit temperature with varying fresh gas flow. Auxiliary temperature sensors provided information on air temperatures in the warming chamber. The heating unit was placed between the distal end of the anesthetic circuit and the entry port to the warming chamber. Due to heat losses from the warming chamber itself, the temperature of the incoming gas was higher than the target patient temperature.

### PW1% group

The warming chamber was heated for 35 minutes before rat entry to achieve a box temperature of 34.4 ± 1.6°C. Chamber heating continued after a rat was introduced to the chamber until core temperature increased by 1% over baseline for each animal. Once the target temperature was achieved, general anesthesia was induced with 5% isoflurane in oxygen at 1L/min. At loss of righting reflex, the rat was removed from the warming chamber and placed on an absorbent pad (17" x 24", Ultra Blok, A.M.G. Medical Inc. Montreal, QC) with no further active heat source and core temperature allowed to decrease to below the hypothermic threshold.

### PW40 group

The methods was as described for the PW1% group, except that the target core temperature was 40.0°C before beginning general anesthesia.

### NW group

Rats were placed in the warming chamber for 10 minutes (based on the initial PW1% experiment time within warming chamber) with the same oxygen flow rate as during the PW1% and PW40 treatments. No heat was provided and anesthesia maintained until core temperature decreased below the hypothermic threshold.

### Statistical analysis

Data were analysed with commercial software (Prism 8.1.2, GraphPad Software, La Jolla, CA, USA and MedCalc Software 18.5, Ostend, Belgium). All data approximated a normal distribution according to the D'Agostino-Pearson Omnibus normality test. Time to hypothermia (individualised to each rat) was assessed with a repeated measures 1-way ANOVA (post-hoc Tukey). The effectiveness of the different treatments were assessed with an area under the curve. Curve limits were set as start of anesthesia (time 0) and onset of hypothermia. The area under the curve was assessed with a 1-way ANOVA (post-hoc Tukey test). Agreement between different temperature measurement sites and core temperature were evaluated with a Bland-Altman analysis for repeated measures with data pooled from the three treatment groups. The criterion method (core temperature) was subtracted from the other measurements (tail, fur and rectal). Differences between treatment groups in the percentage of time spent facing the heat source was assessed with a mixed-effect analysis (post-hoc Tukey test). p-values of < 0.05 were considered significant. Data are presented as mean ± SD in the text and mean ± SEM or median ± 10–90 percentile in the figures. Data supporting the results are available in an electronic repository: Pang, Daniel, 2020, "Pre-warming and perianesthetic hypothermia_01", https://doi.org/10.7910/DVN/GCIXJ3, Harvard Dataverse, V1, UNF:6:KaxaaRzSaksvTc+-8qIe18Q = = [fileUNF].

## Results

None of the pre-established exclusion criteria was met during the experiment and each rat completed all experiments (n = 17 / group). Nine video files were corrupted and therefore, behavioural observations could not be completed for these animals (PW1%: n = 8, PW40: n = 1). Additionally, one rat from the PW1% group was excluded from behavioural analysis as an outlier: its behavior differed from all other animals in spending 83% of the time facing the heat source.

The mean core temperature of all rats during baseline (day before experimentation) was 37.2 ± 0.17˚C. Therefore, an overall hypothermia threshold value of 36.9˚C. Pre-warming was successful in increasing core temperature while in the warming chamber. The time to increase core temperature by 1% or to 40˚C was 11 ± 5.1 and 23 ± 5.3 minutes, respectively. The mean core temperatures of the PW1%, PW40 and NW groups at time 0 were 38.5 ± 0.6, 39.6 ± 0.2 and 37.9 ± 0.4˚C, respectively. Pre-warming was also effective in raising skin temperature, with an increase from baseline of 4.8 ± 1.6˚C for the PW1% group and 3.4 ± 1.2˚C for the PW40˚C group. The NW group had a small increase in skin temperature when placed in the warming chamber of 1.6˚C ± 1.1˚C. The core body temperatures of all rats reduced when warming was stopped, and anesthesia started (Fig 1).

## Area under the curve during temperature reduction

Increasing core temperature before the onset of general anesthesia was associated with a significantly greater area under the curve until hypothermia was reached. Significant differences

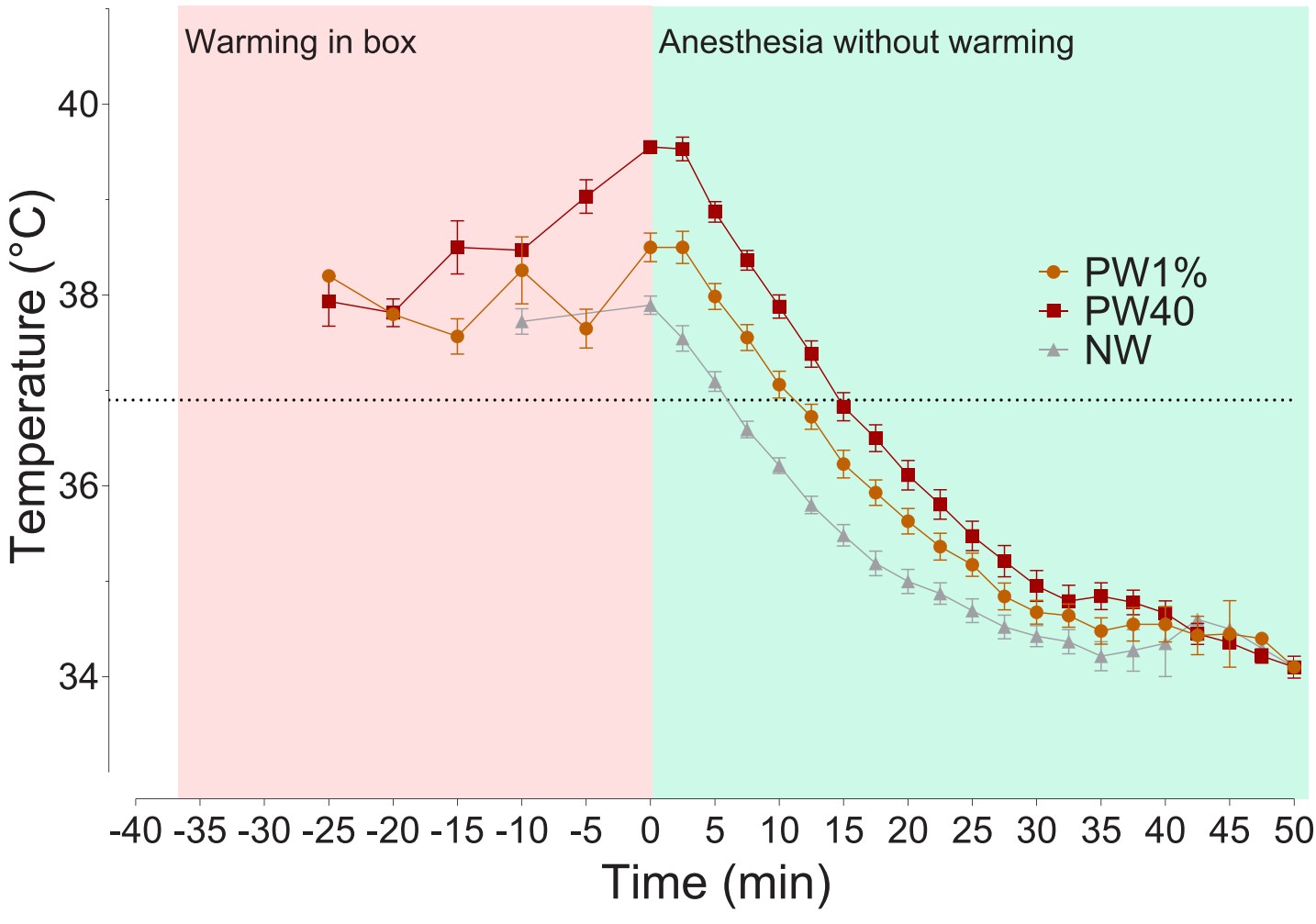

**Fig 1. Core temperature changes in rats pre-warmed to 40˚C (PW40, n = 17), 1% above baseline temperature (PW1%, n = 17) or without warming (NW, n = 17).** Time spent in warming chamber is highlighted in red (time -35 to 0 mins), followed by induction of general anesthesia and removal from warming chamber (green box, time 0 to 50 mins). Area under the curve during temperature reduction was calculated from time 0 to 50 mins. Before time 0, data are plotted every 5 minutes for clarity. Data presented as mean ± SEM.

between areas under the curve was found between all treatment groups: PW1% vs NW (p = 0.0108, 95% CI of difference, -14 to -1.4), PW1% to PW 40 (p < 0.0001, -17 to -5.7) and PW40 to NW (p < 0.0001, -25 to -13; Fig 1).

### Time to reach individual hypothermia threshold

Pre-warming had a significant effect on the time to reach the hypothermia threshold. Times to reach the hypothermia threshold were 7.1, 12.4 and 19.3 minutes for treatment groups NW, PW1% and PW40, respectively. These times were significantly different between each treatment group: PW1% *versus* PW40 (p = 0.004, 95%CI -12 to -2.2), PW40 *versus* NW (p < 0.0001, 95%CI 8.1 to 16), PW1% *versus* NW (p = 0.003, 95%CI 1.8 to 8.7, Fig 2).

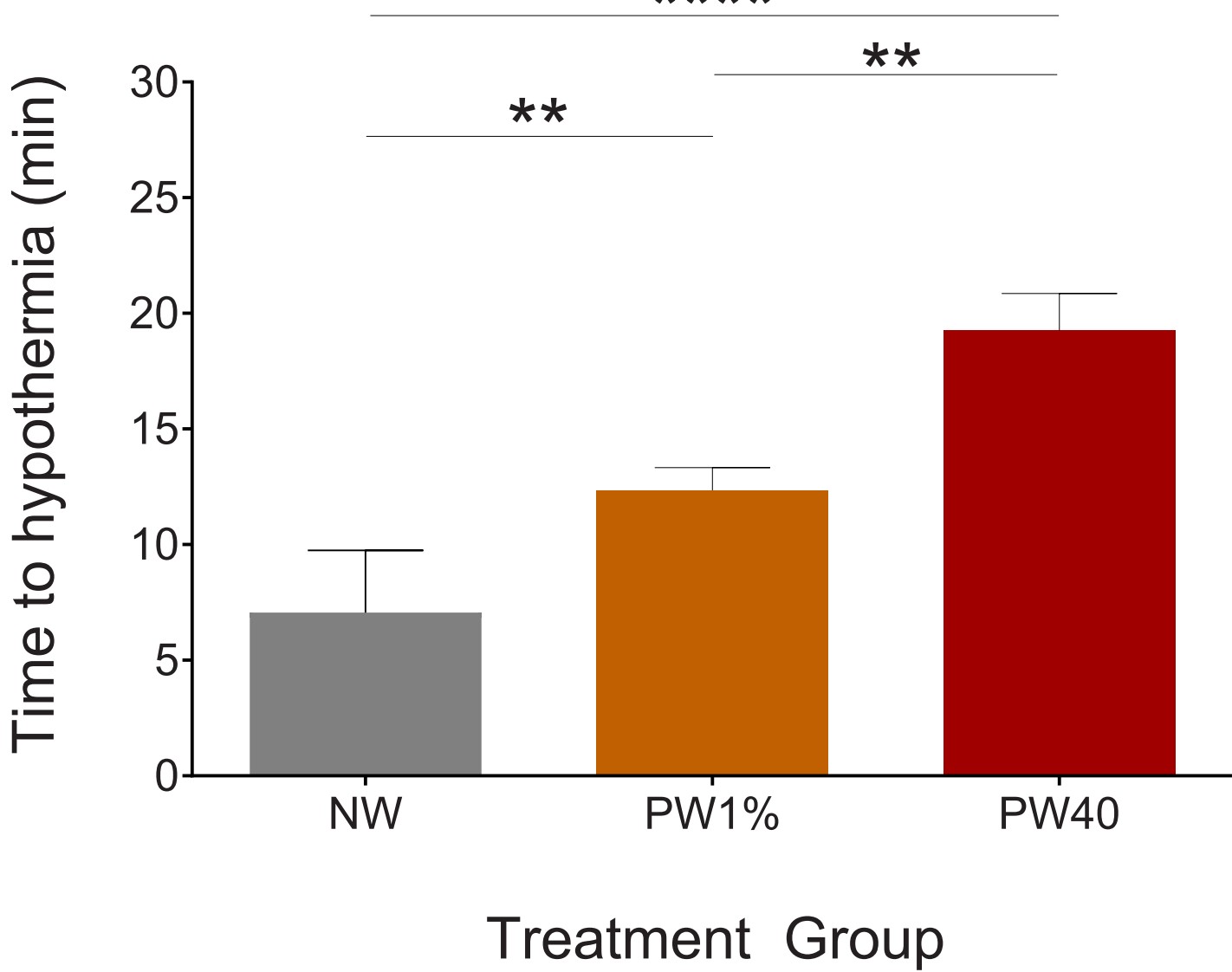

**Fig 2. Time to reach individual hypothermia threshold.** The no-warming (NW, n = 17) group reached their individual hypothermia threshold more quickly than the pre-warming to 1% above baseline core temperature (PW1%, n = 17) (p < 0.01) and pre-warming to 40°C (PW40, n = 17) (p < 0.0001) groups. The PW1% group reached their individual hypothermia threshold more quickly than the PW40 group (p < 0.01). Data presented as mean ± SEM. **p < 0.01. ****p < 0.0001.

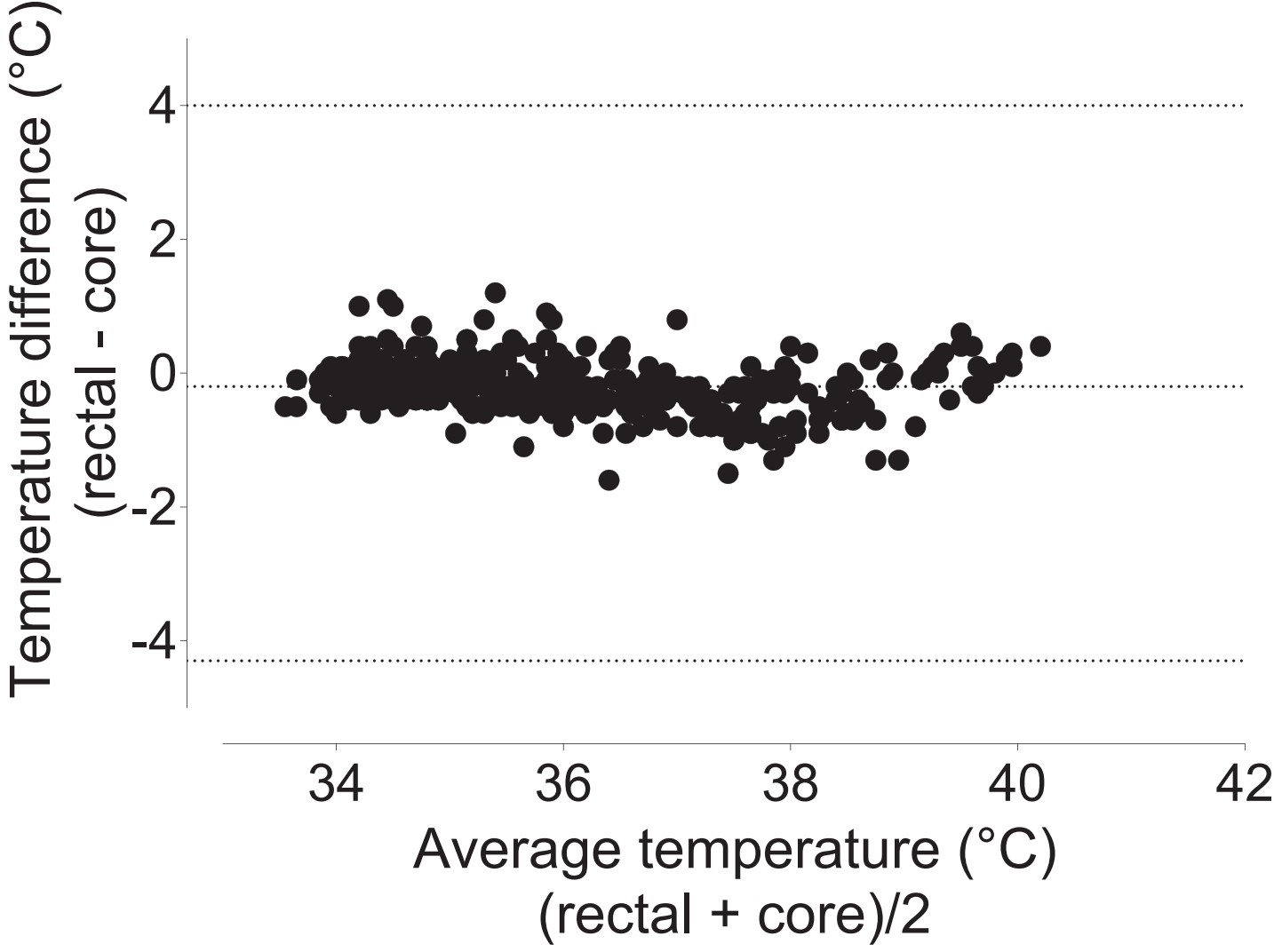

**Fig 3. Bland-Altman plot of repeated measures comparing rectal and core temperatures.** Rectal temperature underestimates core temperatures by 0.20˚C, with 95% limits of agreement ranging from -4.3 to 4.0. Data were pooled from the three treatment groups.

## Comparisons of different temperature measurements

Rectal temperatures approximated core temperatures (bias -0. 20˚C, 95% limits of agreement -4.3 to 4.0; Fig 3). Fur and tail temperatures underestimated core temperatures (fur: bias -2.5˚C, 95% limits of agreement -6.6 to 1.7 Fig 4; tail: bias -7.8˚C, 95% limits of agreement -15.3 to -0.30; Fig 5).

## Behavioural assessment

During both timepoints evaluated, rats that were prewarmed preferred to face away from the heat source during both three-minute intervals (Fig 6). During the first and last three minutes of observation, NW animals did not display a position preference and were more likely to face the heat source in comparison to PW1% (first 3 mins: p = 0.0005, 95%CI 28.3 to 68.1; last 3 mins: p = 0.009, 95%CI 15.1 to 80.5) and PW40 (first 3 mins: p = 0.003, 95%CI 11.2 to 53.5; last 3 mins: p = 0.016, 95%CI 6.5 to 63.2). During the first three minutes, PW40 animals were

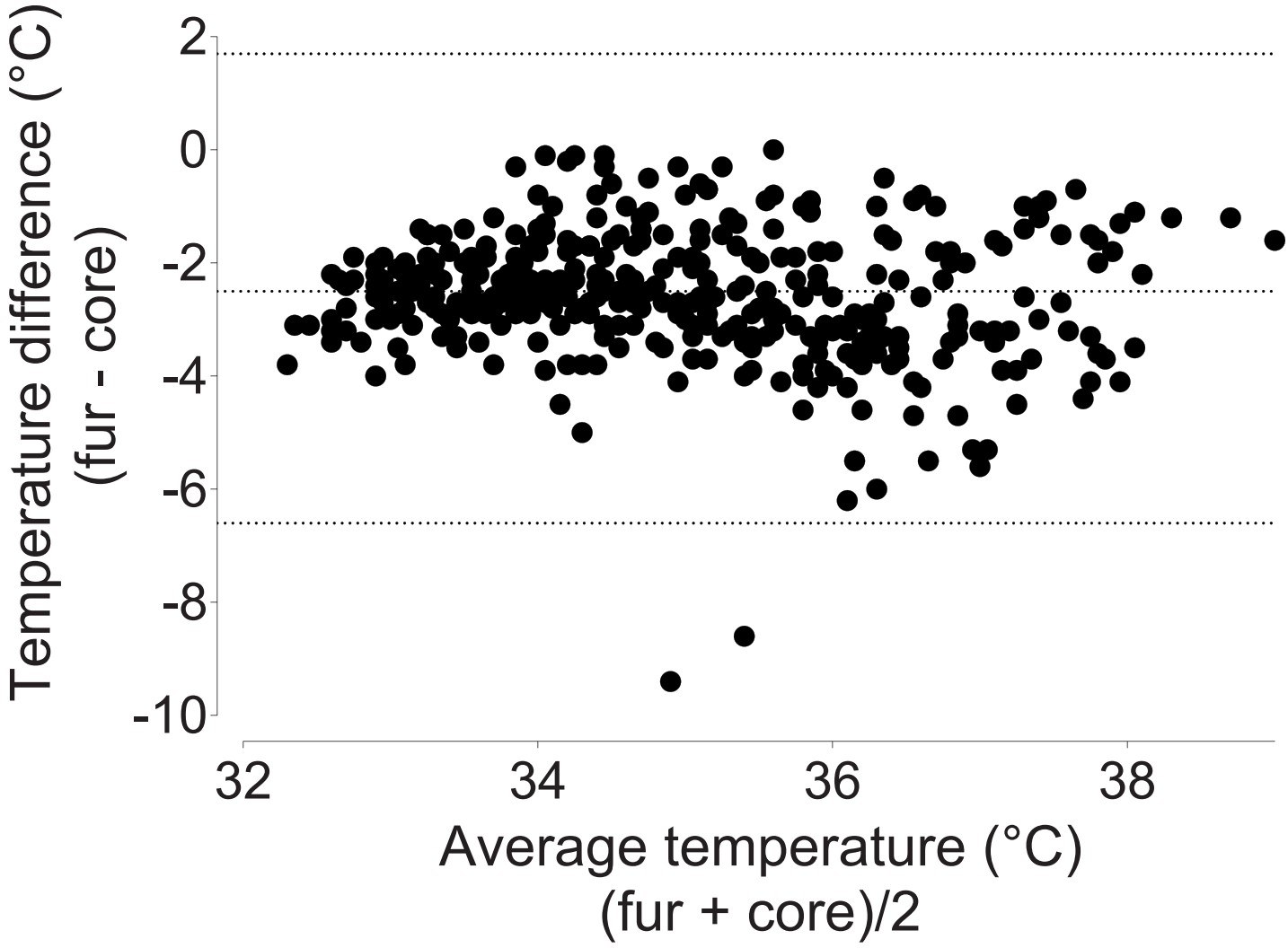

**Fig 4. Bland-Altman plot of repeated measures comping fur and core temperatures.** Fur temperature underestimates core temperature by 2.50˚C, with 95% limits of agreement ranging from -6.7 to 1.7. Data were pooled from the three treatment groups.

more likely to face the heat source than PW1% animals (p = 0.016, 95%CI 3.59 to 28.2). A low incidence of digging behaviour was observed in both PW1% (n = 2/9) and PW40 (n = 3/16) groups, which was only displayed during the second observation period (end of warming period). Chromodacryhorrhea was observed in two rats that displayed digging behavior in the PW40 group. Neither digging behaviour nor the occurrence of chromodacryhorrhea were evident in the NW group. No other abnormal behaviours were observed.

## Discussion

The main findings of this study are that pre-warming is effective at delaying the onset of hypothermia during general anesthesia. This was achieved by increasing skin and core temperatures. Following pre-warming, the rate of temperature loss was slightly faster than without pre-warming. Additionally, the accuracy and agreement of tail and fur temperature, as proxies of core temperature, was poor, whereas rectal temperature showed good agreement with core temperature.

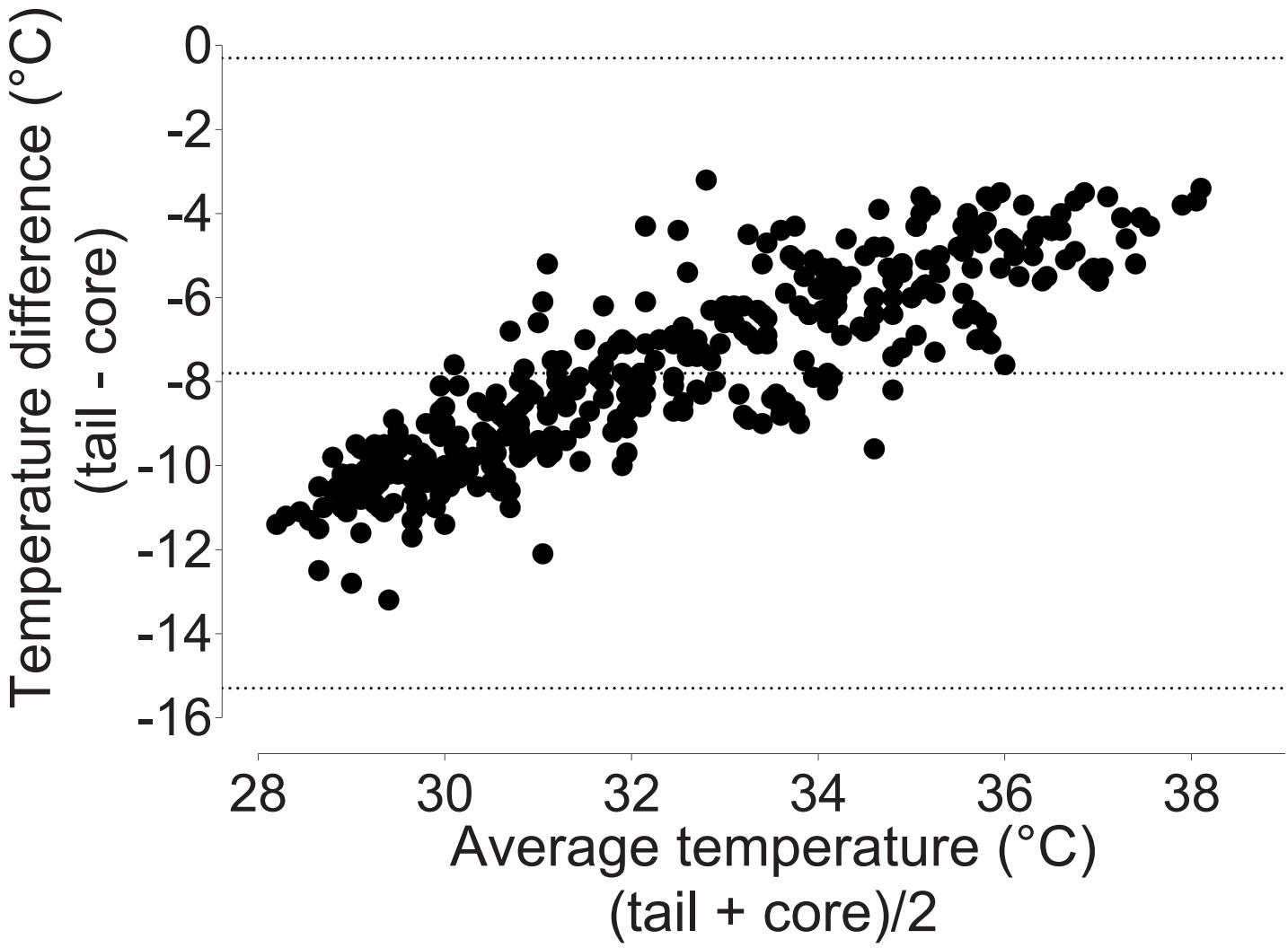

**Fig 5. Bland-Altman plot of repeated measures comparing tail and core temperatures.** Tail temperature underestimates core temperature by -7.80˚C, with 95% limits of agreement ranging from -15.3 to -0.3. Data were pooled from the three treatment groups.

The consequences of hypothermia are well documented in human medicine, with a decrease in core temperature of 1˚C linked to significant adverse outcomes. These include increased surgical site infection, hemorrhage, impaired immune function, thermal discomfort and prolonged recovery and hospitalisation [11, 23–28]. In laboratory mice, warming animals between injection of anesthetic agents and induction of anesthesia resulted in reduced data variability [29].

Despite these known adverse effects, hypothermia remains a common peri-anesthetic complication in both human and veterinary medicine. Recent studies have documented incidence rates as high as 84–97% in cats and dogs undergoing a variety of procedures [1, 3, 4]. Good temperature management is a key element in optimal recovery from surgery and is included in the concept of Enhanced Recovery After Surgery (ERAS), a perioperative management strategy applied in human medicine to optimise recovery (return to normal function) without compromising pain management [30, 31]. ERAS is in its infancy in veterinary medicine [32–34].

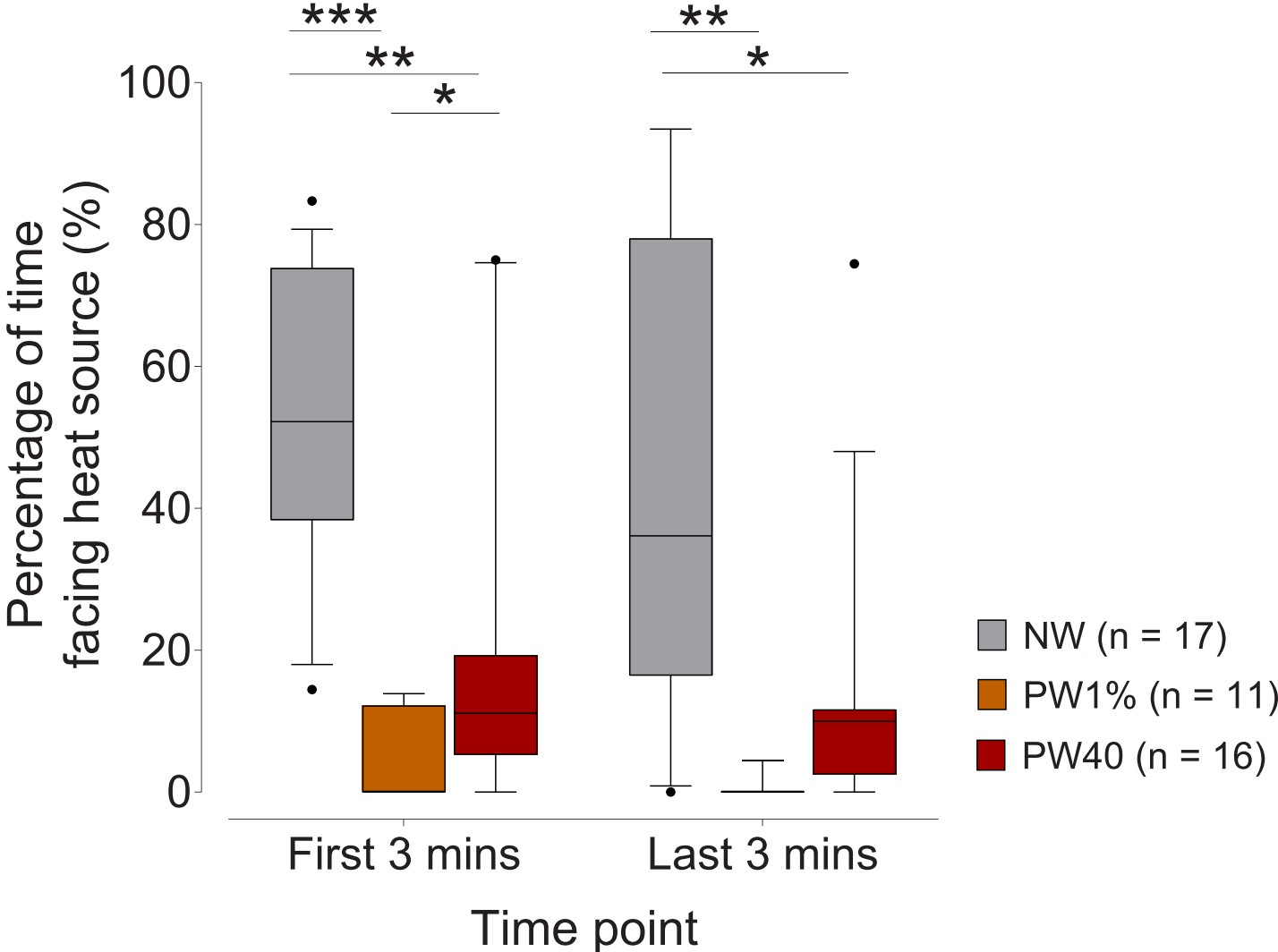

**Fig 6. Percentage of time rats faced the heat source.** The no-warming (NW, n = 17) group spent approximately 50% of the time facing the heat source and this was significantly longer than the pre-warming to 1% above baseline core temperature (PW1%, n = 17) and pre-warming to 40˚C (PW40, n = 17) groups during the first and last 3 mins of observation (p < 0.05). During the first three minutes, the PW40 group was more likely to face the heat source than PW1% (p < 0.05). Data presented as median ± 10–90 percentile. *p < 0.05. **p < 0.01. ***p < 0.001.

In mammals, core body temperature is normally closely regulated by the hypothalamus, maintaining core temperature within ± 0.3˚C through various autonomic and behavioral mechanisms. General anesthesia (injectable and volatile anesthetics) prevents heat-seeking behaviors, inhibits heat-producing activities (i.e. shivering) and loosens the regulation of core temperature so that fluctuations in core temperature of 3–6˚C are permitted. Protective strategies against hypothermia are impaired: vasoconstriction does not occur until a lower temperature is attained and there is a loss of control over arteriovenous shunting [15, 35, 36].

Inhibition of thermoregulation promotes a major redistribution of heat from the core to the periphery, explaining the rapid drop in core temperature noted during the first hour of general anesthesia [7]. This rapid onset of hypothermia is well documented in both rats and mice [14, 20, 37].

The concept of pre-warming patients was introduced in humans on the basis that increasing peripheral temperature before general anesthesia would limit temperature redistribution from the core to the periphery, subsequently delaying hypothermia [18]. This was successfully achieved by raising skin temperatures by approximately 4 to 5˚C, an increase associated with small increases in core temperature (0.3–0.5˚C) [19]. When coupled with intra-operative warming, this strategy was effective at preventing hypothermia [38–40].

The literature on peri-anesthetic temperature management in rodents is limited, with a focus on warming after general anesthesia is induced [37, 41, 42]. However, recent preliminary work has shown that pre-warming can be effective. Warming an anesthetic induction box before and during induction of general anesthesia with isoflurane in rats (box temperature 35.7 ± 3.5˚C to 37.5 ± 2.6˚C) was successful in maintaining core temperature above baseline during 40 minutes of general anesthesia in conjunction with active warming using a heat pad (set at 40˚C); however, this study did not investigate the effects of pre-warming in isolation [20].

The PW1% group was determined based on human literature, which shows an increase in core temperature of approximately 1% above baseline to be effective in preventing core to periphery heat redistribution [19, 40]. The PW40 treatment was selected to assess the effect of warming to a higher core temperature on behavior and temperature maintenance. An upper limit of 40˚C was selected as core temperatures in mice of 41.5 ± 0.1˚C for 2 hours resulted in apoptosis [43].

Pre-warming was successful in delaying the onset of hypothermia, which took approximately 2.5 times longer to occur in the PW40 group and 1.5 times longer in the PW1% group compared with the NW group. Overall, pre-warming conferred protection against hypothermia for approximately 15 minutes without additional warming, a duration suitable for short procedures. Beyond this period, normothermia could be simply maintained with active warming and appropriate warming during recovery, as previously shown [20, 44]. As has been reported in humans, the rate of heat loss following induction of anesthesia was greater than in the pre-warmed than in the NW animals, reflecting the temperature gradient to the environment [9, 45].

The use of telemetry capsules placed within the peritoneal cavity has the advantages of accurately reflecting core temperature and allowing remote monitoring [46]. However, it is clearly not a practical approach for routine monitoring. Therefore, several other temperatures were recorded for comparison against core temperature. Rectal temperature performed well, with values acceptably close to core temperature. This is in line with previous work showing a similar bias and limits of agreement between rectal and core temperatures: Bias; -0.9˚C, limits of agreements; 0.1 to -1.9˚C [20]. Importantly, rectal temperature accuracy is dependent on thermometer insertion depth [47]. In contrast to rectal temperature, fur and tail temperature performed poorly. Comparing these alternative sites for temperature measurement with telemetric core temperature allows an accurate determination of their inter-relationship. These findings show that rectal temperature can be used as a proxy for core temperature in healthy rats but fur and tail temperatures are highly variable and unlikely to be useful in estimating core body temperature. The application of the Bland-Altman method reveals this variability, in contrast to interpreting correlation alone [48].

The behavioral analyses were equivocal. Though rats in the pre-warmed groups showed a clear preference for facing away from the heat source, overt signs of distress were rarely displayed. Nonetheless, further investigation is required to characterise the presence and severity of stress. A larger warming chamber that would allow rats to move away from the heat source may be preferable.

## Limitations

This study had several limitations. The study was limited to a single volatile anesthetic; therefore, the role of pre-warming during injectable anesthetic drug use remain unknown. There was no measurement of air humidity or velocity in the warming chamber; both factors affect heat loss through evaporation [49]. Any adverse effects of the transient increase in core temperature to 40°C are unknown as neurological outcomes were not studied following recovery; however, no overt changes in behavior or body condition were noted in the weeks following the study. Finally, the study design was limited to a simple procedure performed in healthy animals. While the general principles are unlikely to change, the rate and degree of heat loss during invasive procedures (entering a body cavity) or in systemically sick animals are likely to differ.

## Conclusion

Pre-warming alone is effective in delaying hypothermia in rats anesthetized with isoflurane. The duration of effect was short, necessitating temperature support for longer anesthetic periods. Rectal temperature measurement is an acceptable proxy for core temperature, unlike fur and tail temperature. Further research is needed to establish temperature profiles and optimal temperature management during surgery and in sick animals.

## Acknowledgments

The authors thank Dr Guy Beauchamp (Université de Montréal) for statistical advice.

## Author Contributions

**Conceptualization:** Daniel S. J. Pang.

**Data curation:** Maxime Rufiange, Daniel S. J. Pang.

**Formal analysis:** Maxime Rufiange, Vivian S. Y. Leung, Daniel S. J. Pang.

**Funding acquisition:** Daniel S. J. Pang.

**Investigation:** Maxime Rufiange.

**Methodology:** Maxime Rufiange, Keith Simpson, Daniel S. J. Pang.

**Project administration:** Daniel S. J. Pang.

**Resources:** Daniel S. J. Pang.

**Supervision:** Daniel S. J. Pang.

**Visualization:** Maxime Rufiange, Vivian S. Y. Leung, Daniel S. J. Pang.

**Writing – original draft:** Daniel S. J. Pang.

**Writing – review & editing:** Daniel S. J. Pang.

**Writing – original draft:** Maxime Rufiange, Vivian S. Y. Leung.

**Writing – review & editing:** Maxime Rufiange, Vivian S. Y. Leung, Keith Simpson.

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
