## [Decision Letter · Decision Letter 0]

22 Aug 2019

PONE-D-19-18070

Pre-warming before general anesthesia with isoflurane delays the onset of hypothermia in rats

PLOS ONE

Dear Dr. Pang,

Thank you for submitting your manuscript to PLOS ONE. After careful consideration, we have decided that your manuscript does not meet our criteria for publication and must therefore be rejected.

Specifically:

The authors investigated theeffect of pre-warming at delaying the onset of hypothermia during general anesthesia. They compared different pre-warming temperature regimens and performed a comprehensive analysis. However, there were enormous concerns regarding the current manuscript.

1. Reviewers and I all especially concerned about the novelty of the article.

2. The effectiveness and clinical implicationof pre-warming treatment could not be ensured.Specially, the protection of pre-warming against hypothermia was no more valuable, because in clinical practice, the duration of anesthesia is often longer than 15 minutes or even hours. In addition, the core temperature is greatly affected by numerous environmental factors, not just general anesthetics.

3. The safety of pre-warming treatment was another important issue. The upper limit of 40°C was selected based on the results that the core temperatures in mice of 41.5 ± 0.1°C for 2 hours could resulted in apoptosis. However, it was not yet proven whether 40°C has a long-term effect on brain function. It is essential to assess intraoperative and postoperative neurobehavioral outcomes.

Unfortunately, the currentmanuscript could not give the readers more valuable clinical significance and evidence-based conclusions. 

I am sorry that we cannot be more positive on this occasion, but hope that you appreciate the reasons for this decision.

Yours sincerely,

JianJun Yang, M.D., Ph.D.

Academic Editor

PLOS ONE

Reviewers' comments:

Reviewer's Responses to Questions

**Comments to the Author**

1. Is the manuscript technically sound, and do the data support the conclusions?

Reviewer #1: Yes

Reviewer #2: Partly

2. Has the statistical analysis been performed appropriately and rigorously? 

Reviewer #1: Yes

Reviewer #2: No

3. Have the authors made all data underlying the findings in their manuscript fully available?

Reviewer #1: Yes

Reviewer #2: Yes

4. Is the manuscript presented in an intelligible fashion and written in standard English?

Reviewer #1: Yes

Reviewer #2: Yes

5. Review Comments to the Author

Reviewer #1: The authors aimed to investigate the effect of prewarming on the isoflurane-induced hypothermia in the rat. It was demonstrated that prewarming confers a protective effect against hypothermia during volatile anesthesia. The animal experiment seems to have been adequately performed. However, it is not clear what is the novel finding of this study.

1. Figures 1 and 2 seem to show the same data. How was the data in Figure 2 obtained? The authors should explain why Figure 2, in addition to Figure 1, is necessary in this paper.

2. Figure 3 is obtained from Figure 1. It is not clear why Figure 3 is necessary in this paper.

3. Effectiveness of prewarming to maintain core temperature during anesthesia has been already reported (lines 371-376). The present study shows the effect of prewarming in more detail, but novelty compared with the previous report is not clear. The author should more clearly explain novelty of this research.

4. Differences among core, rectal and fur temperature are as expected (Figures 4-6). What is the novel finding of this data?

5. It is not clear why the behavioral analyses were necessary and what was suggested by the findings of the behavioral analyses.

Reviewer #2: PONE-D-19-18070

In this preclinical study, the Authors evaluated the prewarming animals before induction of general anesthesia would delay the onset of hypothermia. A secondary objective was to compare the accuracy of different temperature measurement sites to core temperature (telemetric capsules implanted in the abdomen).

This prospective, crossover study (n = 17 adult male and female SD rats) compared three treatment groups: PW1% (pre-warming to increase core temperature 1% over baseline), PW40 (pre-warming to increase core temperature to 40°C) and NW (no warming).

The PW1% group was completed first to ensure tolerance of pre-warming.

Treatment order was then randomized and alternated after a washout period. Once target temperature was achieved, anesthesia was induced and maintained with isoflurane in oxygen without further external temperature support.

Pre-warming was effective at delaying the onset of hypothermia, with a significant difference between PW1% (11.2 minutes) and PW40 (14.7 minutes, p = 0.0044 (95%CI -12 to -2.2), PW40 and NW (6.0 minutes, p = 0.003 (95%CI 1.8 to 8.7) and PW1% and PW40 (p = 0.004, 95%CI -12 to -2.2).

The rate of heat loss in the pre-warmed groups exceed that of the NW group: PW1% versus NW (p = 0.005, 95%CI 0.004 to 0.027), PW40 versus NW (p < 0.0001, 95%CI 0.014 to 0.036) and PW1% versus PW40 (p = 0.07, 95%CI -0.021 to 0.00066).

The Authors concluded that, pre-warming alone confers a protective effect against hypothermia during volatile anesthesia; however, longer duration procedures would require additional heating support.

Comments

The study confirms in various animal subset an information already proven in humons.

Not having a sample size calculation, it is not clear the strength of reported results and derived conclusions

I would suggest to summarize the overall meaning of reported results in the first sentence of the Discussion section.

Discussion should be shortened (by 10 to 15% or more) and focused on clinical implication of the present study.

Please edit the Reference section according the journal requirements

6. PLOS authors have the option to publish the peer review history of their article (what does this mean?). If published, this will include your full peer review and any attached files.

Reviewer #1: No

Reviewer #2: No

- - - - -

---

## [Author Response · Author response to Decision Letter 0]

2 Oct 2019

Please see uploaded Cover letter and Author response documents.

---

## [Decision Letter · Decision Letter 1]

6 Jan 2020

PONE-D-19-18070R1

Pre-warming before general anesthesia with isoflurane delays the onset of hypothermia in rats

PLOS ONE

Dear Dr Pang,

Thank you for submitting your manuscript to PLOS ONE. Following your appeal to the initial decision to reject, your manuscript has now been assessed by two independent editors, and has been reviewed by an additional peer reviewer. As you will see, the third peer reviewer is generally positive, but has made some suggestions for how to improve the ms prior to final publication. Therefore, we invite you to submit a revised version of the manuscript that addresses the points raised during the review process.

We would appreciate receiving your revised manuscript by Feb 20 2020 11:59PM. To enhance the reproducibility of your results, we recommend that if applicable you deposit your laboratory protocols in protocols.io, where a protocol can be assigned its own identifier (DOI) such that it can be cited independently in the future. For instructions see: http://journals.plos.org/plosone/s/submission-guidelines#loc-laboratory-protocols

We look forward to receiving your revised manuscript.

Kind regards,

Matthew Parker

Christopher James Johnson, Ph.D.

Academic Editors

PLOS ONE

Journal Requirements:

'Natural Sciences and Engineering Research Council (NSERC) Discovery Grant (ID:

424022-2013; DSJP), Fondation Lévesque (DSJP). The funders had no role in study

design, data collection and analysis, decision to publish, or preparation of the

manuscript.'

We note that one or more of the authors are employed by a commercial company: 3Vetronic Services Ltd.

Reviewers' comments:

Reviewer's Responses to Questions

**Comments to the Author**

1. If the authors have adequately addressed your comments raised in a previous round of review and you feel that this manuscript is now acceptable for publication, you may indicate that here to bypass the “Comments to the Author” section, enter your conflict of interest statement in the “Confidential to Editor” section, and submit your "Accept" recommendation.

Reviewer #3: (No Response)

2. Is the manuscript technically sound, and do the data support the conclusions?

Reviewer #3: Yes

3. Has the statistical analysis been performed appropriately and rigorously? 

Reviewer #3: Yes

4. Have the authors made all data underlying the findings in their manuscript fully available?

Reviewer #3: Yes

5. Is the manuscript presented in an intelligible fashion and written in standard English?

Reviewer #3: No

6. Review Comments to the Author

Reviewer #3: This is a useful study, addressing an issue that is of importance when anaesthetising both human and veterinary subjects. It is unfortunate that little emphasis is given to the important implications for research animal anaesthesia - hypothermia has the potential to influence numerous animal models, the effects may last well beyond the period of anaesthesia, and hypothermia occurs commonly when anaesthetising rodents. Although this effects is more common in mice, it is also relevant in rats, and in any event the effects noted in rats are of relevance to mice.

I have one specific comment:

Redistribution of warm blood from the core to the periphery is the primary

24 mechanism in the development of hypothermia and begins following induction of

25 anesthesia.

This is correct when the anaesthetic agent produces peripheral vasodilation, but the most widely used agent for rodents, the combination of ketamin and xylazine, produces a peripheral vasoconstriction. Perhaps it might be better to say that “when using inhalational anaesthetic agents, ….

There are a small number of typographical errors that could be corrected with further proof-reading of the manuscript.

7. PLOS authors have the option to publish the peer review history of their article (what does this mean?). If published, this will include your full peer review and any attached files.

Reviewer #3: Yes: Paul Flecknell

---

## [Author Response · Author response to Decision Letter 1]

9 Jan 2020

Please see uploaded document, "response to reviewers_PONE-D-19-18070R1".

---

## [Editor Report · Decision Letter 2]

22 Jan 2020

Pre-warming before general anesthesia with isoflurane delays the onset of hypothermia in rats

PONE-D-19-18070R2

Dear Dr. Pang,

We are pleased to inform you that your manuscript has been judged scientifically suitable for publication and will be formally accepted for publication once it complies with all outstanding technical requirements.

With kind regards,

Matthew Parker

Academic Editor

PLOS ONE
---

## [Editor Report · Acceptance letter]

13 Feb 2020

PONE-D-19-18070R2 

Pre-warming before general anesthesia with isoflurane delays the onset of hypothermia in rats 

Dear Dr. Pang:

I am pleased to inform you that your manuscript has been deemed suitable for publication in PLOS ONE. Congratulations! Your manuscript is now with our production department. 

With kind regards,

on behalf of

Dr. Matthew Parker 

Academic Editor

PLOS ONE